# The Root Nodule Microbiome of Cultivated and Wild Halophytic Legumes Showed Similar Diversity but Distinct Community Structure in Yellow River Delta Saline Soils

**DOI:** 10.3390/microorganisms8020207

**Published:** 2020-02-03

**Authors:** Yanfen Zheng, Jing Liang, Dong-Lin Zhao, Chen Meng, Zong-Chang Xu, Zhi-Hong Xie, Cheng-Sheng Zhang

**Affiliations:** 1Marine Agriculture Research Center, Tobacco Research Institute of Chinese Academy of Agricultural Sciences, Qingdao 266101, China; 2Special Crops Research Center of Chinese Academy of Agricultural Sciences, Qingdao 266101, China; 3Yantai Institute of Coastal Zone Research, Chinese Academy of Sciences, Yantai 264003, China

**Keywords:** nodule microbiome, halophytic legume, *Sesbania cannabina*, *Glycine soja*, 16S rDNA full-length amplicon sequencing

## Abstract

Symbiotic associations between leguminous plants and their nodule microbiome play a key role in sustainable agriculture by facilitating the fixation of atmospheric nitrogen and enhancing plant stress resistance. This study aimed to decipher the root nodule microbiome of two halophytic legumes, *Sesbania cannabina* and *Glycine soja*, which grow in saline soils of the Yellow River Delta, China, using PacBio’s circular consensus sequencing for full-length bacterial 16S rRNA gene to obtain finer taxonomic information. The cultivated legume *Glycine max* was used for comparison. We identified 18 bacterial genera and 55 species in nodule samples, which mainly classified to *Proteobacteria*, and rhizobial genus *Ensifer* was the predominant group. The three legumes showed similarity in operational taxonomic unit (OTU) diversity but distinction in OTU richness, indicating that they harbor similar bacterial species with different relative contents. The results of principal coordinates analysis and ANOSIM tests indicated that *G. soja* and *G. max* have similar nodule bacterial communities, and these communities differ from that of *S. cannabina*. Wild legumes *S. cannabina* and *G. soja* both harbored a higher number of rhizobia, while *G. max* possessed more non-rhizobial bacteria. These differences could be associated with their adaptability to saline–alkali stress and revealed clues on the nodule endophytes with relative importance of culturable rhizobial symbionts.

## 1. Introduction

Soil salinization is one of the major abiotic factors reducing crop productivity worldwide, and it is estimated that there is currently approximately one billion hm^2^ of salt-affected soils distributed across more than 100 countries [1]. The problem is particularly acute in China, which has a large area (36,300,000 hm^2^) and wide distribution of salinized soil [2]. The Yellow River Delta located in coastal areas of the Yellow River estuary has a total area of 620,000 hm^2^ of saline–alkaline land. As a typical representative of the coastal saline soils of China, the soil of the delta has a high salt content, low organic matter content, and is nitrogen deficient. However, due to the low groundwater level and frequent seawater intrusion [3], existing saline soil improvement technologies, for example, engineering and physical and chemical improvement, are generally ineffective. 

Studies have shown that utilization of salt-tolerant plants is a new and promising approach for the improvement of saline–alkaline soils. In this regard, symbiotic nitrogen-fixing legume plants, which are characterized by the formation of root nodules, could grow well in saline-affected soils [4]. Moreover, wild legumes were thought to be more tolerant to stress than crop legumes. *Sesbania cannabina* (Retz.) Poir and *Glycine soja* Sieb. et Zucc are annual legumes and included in the list of Halophytes of China [5]. Owing to their characteristic features of high productivity, salt tolerance, and adaptation to nutrient-deficient environments, they have been considered as ideal candidate plants for coastal saline soil improvement.

Rhizobia, which are associated with nodule formation in legume plants, supply nitrogen to the host plants, and can also exist in non-legume plants as beneficial endophytes [6,7]. These rhizobia play important roles in low-input agriculture, land reclamation, and saline soil restoration [8,9,10]. Here, we hypothesized that large amounts of rhizobia were presented in nodules of *S. cannabina* and *G. soja*, which grow naturally in the saline soils of the Yellow River Delta [11], and these two plants could recruit different endophytes of nodules to adapt to saline soils. To date, most studies on *S. cannabina* and *G. soja* have only focused on the biological characteristics and genetic diversity of the culturable rhizobia associated with these plants [12,13], whereas little or no research on the nodule microbiome was conducted. In order to characterize and compare their nodule microbiomes at the species level with high throughput, PacBio’s circular consensus sequencing was performed. This technique could generate near full length 16S rRNA gene and has been validated by Singer et al., [14] and Motooka et al. [15], from a defined mock community. They both found PacBio sequencing could provide more accurate and reliable estimation of microbial communities. In addition, to examine differences between cultivated and wild legumes, we also compared the nodule microbiome of these two species with that of the widely cultivated legume *Glycine max* (Linn.) Merr.

## 2. Materials and Methods 

### 2.1. Nodule Samples

Experiments were conducted at the modern agriculture technology experiment and demonstration base of Shandong Academy of Agricultural Sciences (Dongying) in 2017. The experimental plots used in this study are characterized by moderately saline soil with a salt content of 0.43%. Other chemical properties of the soil are as follows: pH 7.6; organic carbon content, 0.56%; available phosphorus, 4.6 mg/kg; available nitrogen, 64.8 mg/kg; and available potassium, 112 mg/kg. For *G. max,* we used seeds of the variety Qihuang 34, whereas seeds of *S. cannabina* and *G. soja* were collected from wild plants growing in saline soils of the Yellow River Delta. The seeds of all three examined plants were sown on 23 April with amounts of 150, 50, and 100 kg/hm^2^ for *S. cannabina*, *G. max,* and *G. soja,* respectively. Randomized block test with three repetition was used. The planting area of each plot was approximate 222 m^2^. Nodules were sampled on 20 July, when the plants were in the vigorous growth stage. For each plant species, samples were collected at three sampling points, with five plants being collected at random from each point. Entire plant was dug up, packed into self-sealed bags, and immediately transferred to the laboratory under refrigerated conditions.

### 2.2. Observation of Nodulation Characteristics

Nodules were carefully separated from the roots of the collected plants, wrapped within a single layer of gauze, and washed with flowing water and, then, placed in an ultrasonic cleaner (Kun Shan Ultrasonic Instruments Co., Ltd., China) for 2 h to remove the soil adhering to the surface, as well as the surface bacteria. After drying the nodules with filter paper, we recorded the fresh weight, size, and morphology (shape and color) of effective nodules (those with red or pink inner tissue).

### 2.3. Determination of Nitrogenase Activity

The nitrogen fixation activity of nodules was determined using the acetylene reduction method [16]. Briefly, 1 g of mixed nodules was placed to a 10 mL serum bottle (sealed with an isobutyl rubber stopper), into which 2 mL of acetylene gas was injected. After incubation in a water bath (30 °C) for 30 min, 0.1 mL gaseous samples were extracted from the tube using a microinjector and subjected to gas chromatographic analysis (Agilent 7890, USA; column temperature 60 °C, inlet temperature 120 °C, FID monitor temperature 120 °C, gas flow rate 50 mL min^−1^, H_2_ 60 mL min^−1^, and air 50 mL min^−1^).

### 2.4. Microbial DNA Extraction

Nodule samples were surface sterilized in 0.1% HgCl_2_ and, then, thoroughly washed with gentle shaking in magnesium buffer in the presence of 1% Tween 20. Total genomic DNA from nodules (0.5 g for each sample) was extracted according to the instructions of a DNeasy^®^ PowerSoil^®^ Kit (QIAGEN, Germany). This kit used a bead-beating method to lyse cells and can get high DNA yield (60 to 130 ng μL^−1^) though a considerable portion of DNA was from host. Three replicate extractions were performed for each sample. The purity and quantity of the extracted DNA were assessed on 0.8% agarose gels using a Nanodrop (2000) spectrophotometer (Thermo Scientific, USA).

### 2.5. Quantitative PCR (qPCR) of the Bacterial Community

Quantitative PCR was performed to determine the bacterial 16S rRNA gene copy absolute abundances. The extracted nodule DNA was amplified with the primer set Eub338/Eub518 [17]. The qPCR system included 10 μL of 2× SYBR R Premix Ex Taq (Tli RNaseH Plus) (Takara Bio, Dalian, China), 0.4 μL of each primer (10 μM), 0.4 μL of Rox Reference dye II (50×), 2.0 μL of template DNA, and 6.8 μL of dH_2_O. The qPCRs were run at 95 °C for 30 s, 40 cycles at 95 °C for 10 s, 63 °C for 30 s; at the end of each run, melting curves of the PCR products were obtained through cycles performed at 95 °C 3 s, 60 °C 30 s. and 95 °C 3 s. Every qPCR reaction was repeated three times. Standard curves were constructed from a series of dilutions ranging from 100 to 3 × 10^8^ copies/μL. The 16S rRNA gene copy number was calculated with Ct values, and transferred into bacteria number.

### 2.6. 16S rRNA Gene Amplification and PacBio Sequencing

The near full length of the bacterial 16S rRNA gene was amplified from samples using the primers 27F (AGAGTTTGATCCTGGCTCAG) and 1492R (GGTTACCTTGTTACGACTT). For each sample, a 16-digit barcode sequence was added to the 5′ end of the forward and reverse primers (provided by Allwegene Company, Beijing). The PCR was carried out in a Mastercycler Gradient thermal cycler (Eppendorf, Germany) using 25 μL reaction volumes, containing 12.5 μL 2× Taq PCR MasterMix (TIANGEN, China), 3 μL BSA (2 ng/μL), 2 μL primer (5 µM), 2 μL template DNA, and 5.5 μL ddH_2_O. The cycling parameters were as follows: 95 °C for 5 min, followed by 32 cycles of 95 °C for 45 s, 55 °C for 50 s, and 72 °C for 45 s, with a final extension at 72 °C for 10 min. PCR products was mixed in equimolar ratios. Then, mixture PCR products was tested by 2% agarose gel. The target bands were purified using a QIAquick Gel Extraction Kit (QIAGEN, Germany) and quantified using real-time PCR. Sequencing libraries were constructed from five PCR technical replicate products using a PacBio SMRTbell Template Prep Kit (Pacific Biosciences), quantified by Qubit and tested insert size by Agilent 2100. Deep sequencing was performed using the Pacbio RS11 platform by Allwegene Company (Beijing, China).

### 2.7. Sequence Data Analyses

SMRT sequence reads were processed through SMRT Portal to filter sequences for length (<1340 or >1640 bp) and quality. Circular consensus sequences (CCS) making three full passes around the closed-loop amplicon were yielded with high accurate as CCS would effectively reduce random errors. Sequences were further filtered by removing barcode, primer sequences, and sequences if they contained 10 consecutive identical bases. The resulting datasets were then analyzed using usearch (version 8.1). The sequences were clustered into operational taxonomic units (OTUs) at a similarity level of 97% using the UCLUST denovo method to generate rarefaction curves and to calculate richness and diversity indices. The rdp Classifier tool [18] was used to classify representative sequences into different taxonomic groups based on the SILVA SSU reference database (release 128) [19].

To examine the similarity between different samples, principal coordinates analysis (PCoA) were used based on Bray–Curtis distance at the OTU level using PRIMER 6 [20]. Student’s two-tailed t-tests were used to compare morphological characteristics, nitrogenase activity, OTU richness, and diversity indices across different samples. To compare the bacterial communities in different samples, analysis of similarity (ANOSIM) was performed in PRIMER 6 [20]. The complete sequences generated in this study are available in the NCBI SRA database under accession number SRR8163410 (Gs), SRR8163411 (Sc), and SRR8163412 (Gm).

## 3. Results

### 3.1. Natural Field Nodulation

Investigations in the field showed that, whereas the wild legumes *S. cannabina* and *G. soja* grew well, the growth of *G. max* was significantly inhibited by salt stress (Appendix A). Consistent with the observed growth patterns, *G. max* also showed lower nodule weight (Table 1). Almost all nodules were found on the main root and lateral roots; however, we observed certain differences in the shape and size of nodules among the three species, as shown in Table 1. Interestingly, the two wild legumes (*S. cannabina* and *G. soja*) exhibited significantly higher (*P* < 0.05) nodule nitrogenase activity than did *G. max*.

### 3.2. Characteristics of Sample Sequence Tags

A total of 59,023 raw tags (range = 3869 to 11,180, SD = 2301) were obtained by near full-length 16S rRNA gene amplification of nine legume nodule samples using the PacBio platform (Appendix A). After read-quality filtering, a total of 15,134 high-quality sequences were acquired, with 2031 ± 434, 1688 ± 733, and 1325 ± 181 sequences in *G. soja*, *G. max,* and *S. cannabina*, respectively. Normalization was performed using lowest sequencing depth (i.e., 1015 sequences) from one *G. max* nodules sample. On the basis of 97% sequence similarity, the average number of OTUs across all samples was 188 (range = 150 to 264, SD = 34). Rarefaction curves (Figure 1) combined with the estimated coverage values (Table 2), indicated that the libraries were sufficiently large to capture most common OTUs. The highest number of overall unique OTUs was found in the *G. soja* nodules (191 ± 13 per sample), followed by the *G. max* nodules (217 ± 35 per sample) and *S. cannabina* nodules (154 ± 3 per sample). The rarefaction curves also indicated higher numbers of OTUs in *G. soja* and *G. max* nodule samples than that in *S. cannabina* samples. Sequence alignment results showed that most sequences (98.95%, range = 0%–3.75%, SD = 1.17%) could be identified at the species level, whereas only 143 sequences (1.05%) were not assigned to a particular species. 

### 3.3. Microbial Community Richness and Diversity

The richness and diversity values of the bacterial communities revealed no significant difference in bacterial diversity but obvious distinctions in terms of bacterial richness (Table 2). *G. soja* harbored the highest number of OTUs, followed by *G. max*, both of which were significantly higher than the number of OTUs for *S. cannabina*. Similar trends were also observed for Chao1 and PD whole-tree indices. Quantitative PCR revealed that legumes hosted different nodule bacteria numbers (Figure 2). *G. soja* nodules hosted the highest number of cells per gram of sample (5.6 ± 0.68 × 10^5^) as compared with *G. max* (4.2 ± 0.5 × 10^5^) and *S. cannabina* (2.4 ± 0.64 × 10^5^) nodules. These results indicated that although the nodules of the three legumes harbor similar bacterial species, there were clear differences in their relative contents. 

### 3.4. Microbial Taxonomic Analysis at the Phylum and Genus Level

Through a homology comparison with Silva 128 database, all sequences represented by OTUs were identified as belonging to *Proteobacteria* and *Bacteroidetes*. *Proteobacteria* was the core phylum, which was detected in all nodule samples and accounted for 97.64% of total sequences (Figure 3a). Small numbers of *Bacteroidetes* were present in *G. soja* and *G. max*, and, notably, a very few *Acidobacteria* (0.07%) were found in one sample of *G. max*. 

Taxonomic analysis revealed 18 bacterial genera, among which mainly rhizobial genus *Ensifer* group constituted the overwhelming majority in all samples (Figure 3b). Other dominant genera (>0.5%) included *Enterobacter*, *Stenotrophomonas,* and *Chryseobacterium*, which were mainly represented in *G. max* nodules. Of the sequences obtained from nodule samples of *G. soja*, *G. max*, and *S. cannabina*, 85.67% ± 6.29%, 95.36% ± 3.00%, and 99.55% ± 0.12% of sequences were assigned to *Ensifer*. Rhizobial genus *Mesorhizobium* and *Rhizobium* were also detected in these legume nodules.

### 3.5. Microbial Taxonomic Analysis at the Species Levels

Total sequences were classified to 55 species, including rhizobial (31 species) and non-rhizobial (24 species) bacteria. Rhizobia, such as *Ensifer americanum* (0.56% to 46.45%), *Ensifer alkalisoli* YIC4027 (8.10% to 68.23%), and *Ensifer* sp. (13.57% to 17.23%), accounted for most of these sequences in all samples (Table 3). *S. cannabina* harbored highest rhizobia (accounting for 100%), followed by *G. soja* (95.62%) and *G. max* (85.80%). However, the dominant species of each plant species were different. *Ensifer alkalisoli* YIC4027 was dominated in *S. cannabina* (68.23%), while the dominant species in *G. max* was *E. americanum* (42.81%), as did in *G. soja* (46.45%). Small amount of other non-rhizobial bacteria were also detected, which were mainly observed in *G. max* nodules, including *Enterobacter cloacae* (3.62%), *Stenotrophomonas* sp. CanR-75 (2.79%), and *Stenotrophomonas maltophilia* (2.41%).

### 3.6. Comparative Analysis of Bacteria in Different Sample Groups 

A group of 76 OTUs was shared among all three legume nodules (Figure 4a). As shown in Figure 4a, the numbers of OTUs exclusive to *S. cannabina*, *G. soja,* and *G. max* were 6, 24, and 13, respectively, suggesting these three legumes possessed similar nodule microbial species as compared with the large whole OTUs. Sixteen OTUs occurred in 75% of samples (Appendix A), all of which were classified to *Ensifer*. Six OTUs (denovo1901, denovo10033, denovo12569, denovo12913, denovo20293, and denovo27243) were present in all samples of *G. max* and *S. cannabina*, whereas only three OTUs were detected in both *G. soja* and *G. max*. Only two OTUs (denovo29807 and denovo33526), which belonged to *Ensifer* sp. *T2GRs3* and *Ensifer americanum,* respectively, were observed in all nine samples (Appendix A). These two OTUs included 2794 sequences and can be considered as core species in the nodules of the three legumes (Figure 4b). Especially, denovo33526 was dominant in the nodule samples of *G. soja* and *G. max*. 

The results of PCoA indicated that there were differences in the bacterial communities at the OTU level associated with the different nodule samples (Figure 5). Samples from *G. soja* and *G. max* tended to cluster together and were clearly separated from those of *S. cannabina*. The two components extracted explained 81.6% (PC1) and 7.8% (PC2) of data variance. Consistently, on the basis of ANOSIM tests, a significant difference (R = 0.654, *p <* 0.05) was observed among nodule samples from the three legumes (Figure 6), with *S. cannabina* clearly differing from *G. soja* (R = 0.7778, *p* = 0.1) and *G. max* (R = 1, *p* = 0.1). However, no significant difference was observed between *G. soja* and *G. max* (R = -0.222, *p* = 0.8). Collectively, these results indicated a variation in the endophytic nodule bacteria among the legume species. 

## 4. Discussion

In this study, we were able to compile a directory listing the endophytic nodule bacteria at the species level associated with three legumes (*S. cannabina*, *G. soja*, and *G. max*) planted in saline soil based on high-throughput sequencing. Large amounts of rhizobia were found in *S. cannabina* (100%), *G. soja* (95.62%), and *G. max* (85.80%). Nevertheless, different dominant rhizobial species were observed in different plant species. *E. alkalisoli* YIC4027 was the most abundant species in *S. cannabina* (68.23%), which was evidenced to be adaption to saline–alkaline soils [21], while *E. americanum* dominated in *G. soja* (46.45%) and *G. max* (42.81%), indicating these plants could recruit different endophytes of nodules to adapt to saline soils.

In previous studies, only a few genera have been detected in the nodules of different legume species using a cultivation-dependent method [12,13,14]. On the basis of our findings, we could classify total OTUs to one to three phyla and 11 to 53 species across all samples (Appendix A), which extends our knowledge regarding the diversity and richness of the endophytic bacteria associated with legume nodules. At the genus level, *Ensifer* was the major nodule bacteria for all the three legumes (representing 83.72% to 99.69% of total sequences), while only a few sequences were classified to *Rhizobium* and *Mesorhizobium*. These results are consistent with those reported in the previous studies [12,13,14], which identified *Ensifer* as the major nodule rhizobia isolated from *G. soja*, *S. cannabina*, and *G. max,* in China. Other rhizobia have also previously been detected in these three legumes, including *Rhizobium* in *G. soja* [13]; *Agrobacterium*, *Neorhizobium,* and *Rhizobium* in *S. cannabina* [12]; and *Bradyrhizobium* in *G. max* [22]. In addition, Naamala et al. observed that species of *Bradyrhizobium* were the major rhizobia detected in *G. max* [23]. Undoubtedly, fewer rhizobia were characterized in our study, which we speculate could be attributable to the influence of saline soil [24]. This assumption tends to be supported by the result of a previous study that reported rhizobial distribution was affected by environment and soil characteristics [22]. Furthermore, Li et al. found rhizobial populations associated with *S. cannabina* were correlated with soil pH and salinity [12]. The evidence above suggested saline soil in Yellow River Delta could have a significant impact on rhizobia groups and structure in legume nodules. 

In addition to rhizobia, we detected a large number of endophytic non-rhizobial bacteria associated with the nodules of *G. max*. Consistently, Xiao et al. also found large amounts of non-rhizobial genera in *G. max* nodules based on 16S rRNA gene V4-V5 amplicon sequencing [25]. Compared with *G. max*, less abundant non-rhizobial bacteria in *G. soja* nodules was observed in this study. To date, however, there have been no reports describing the non-rhizobial bacteria in *G. soja* nodule. Wu et al. [26] investigated the endophytic bacteria in wild soybean (*G. soja*) root and found *Psedomonas*, *Enterobacter*, *Janthinobacterium*, and *Streptomyces* were the main bacterial genera. In another study, four non-rhizobial genera (*Balneimonas*, *Cronobacter*, *Enterobacter,* and *Pantoea*) were isolated from *S. cannabina* nodules [27], whereas only *Ensifer* was detected in this legume in the present study. Apart from environmental factors, the PacBio CCS technology used in this study for generating near full-length 16S rDNA gene could be an important factor contributing to these observed differences. Recently, this technology has also been used to study microbial community in various samples, for example, Sakinaw Lake water [14], coral [28], and feces [15]. Singer et al. compared PacBio CCS technology with Illumina V4 16S rRNA gene sequences using a defined microbial community and found PacBio CCS technology could generate less ambiguous classification and provide more accurate taxonomic information [14]. Thus, more reliable results could be generated from the sequencing technology we used. 

Although *G. soja*, *G. max,* and *S. cannabina* showed similarities in terms of OTU diversity, there was a difference in OTU richness. Given the close evolutionary relationship between *G. soja* and *G. max*, we not surprisingly detected no significant differences between the OTU richness and diversity of these two legumes, whereas *S. cannabina* had a lower number of OTUs and lower Chao1 richness. In contrast to our observations, Xiao et al. found that the endophytic nodule bacteria associated with *G. max* and *Medicago sativa*, showed similar OTU richness but differed in diversity, indicating that plant species can significantly influence the alpha diversity of endophytic nodule bacteria, as has been reported in previous studies [25,29,30]. *G. soja* and *G. max* also showed similar bacterial community composition, whereas the communities in both these species differed from that in *S. cannabina*, as indicated by the results of PCA analyses (Figure 5) and ANOSIM tests (Figure 6). Bacterial communities in three samples of *G. soja* and *G. max* were not very consistent as shown in Figure 5, which could result from plant individuals or soil heterogeneity, suggesting more samples should be collected in future studies. Importantly, we identified certain OTUs that coexist in two or all three legumes, notably denovo29807 (*Ensifer* sp. *T2GRs3*) and denovo33526 (*E. americanum*) that occurred in all samples, indicating these bacteria can play important roles in nodule microecology. Notably, wild legumes *S. cannabina* and *G. soja* harbored a higher number of rhizobia, while *G. max* possessed diversity of non-rhizobial bacteria. More rhizobia in wild legumes could be associated with their high adaptability to saline–alkali stress.

Recently, increasing attention has focused on the utilization of microbes to mediate plant salt tolerance [31,32,33]. Previous studies have shown that endophytic bacteria associated with halophytic plants invariably show high salt tolerance and play important roles in the salt tolerance of host plants [34,35,36]. Consistently, Rejili et al. reported that rhizobia in the nodules of wild legumes growing in saline soils showed salt tolerance [37]. The fact that some rhizobia are found in the nodules of not only wild halophytic legumes (*G. soja* and *S. cannabina*) but also cultivated legumes (*G. max*) indicates the possibility of utilizing such bacteria to enhance the growth of agricultural crops in saline soils. In this regard, we can envisage the following steps: (1) deciphering the nodule microbiome of halophytic and cultivated legumes to identify the core microbiome; (2) isolating nodule bacterial strains from halophytic legumes according to the core microbiome; and (3) screening for high-efficiency bacterial strains as inoculants that could alleviate salt stress in cultivated legumes. The conceptual strategy for utilizing nodule microbes associated with halophytic legumes is shown in Figure 7; however, further research-based evidence will be necessary to realize the practicality of this strategy. 

## 5. Conclusions

In conclusion, near full-length 16S rRNA gene sequencing analysis revealed 18 bacterial genera from nodule samples of the legume species *G. max*, *G. soja,* and *S. cannabina*. In addition to rhizobia, we also detected a number of other bacterial endophytes associated with the nodules of these legumes. Our results reveal that the root nodule microbiome of cultivated and wild halophytic legumes showed similarity in diversity but distinction in community structure under saline soils. These differences could be associated with their adaptability to saline–alkali stress. This research also provides insights for the future exploitation and utilization of microbial resources associated with wild halophytic legumes.

## Figures and Tables

**Figure 1 microorganisms-08-00207-f001:**
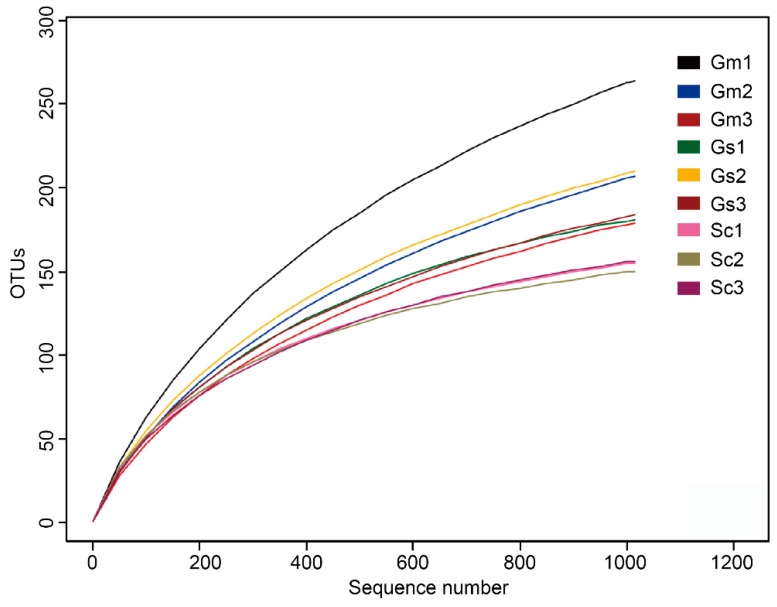
Rarefaction curves based on the 16S rRNA gene with 97% similarity from nodule samples of *Sesbania cannabina* (Sc), *Glycine soja* (Gs), and *Glycine max* (Gm).

**Figure 2 microorganisms-08-00207-f002:**
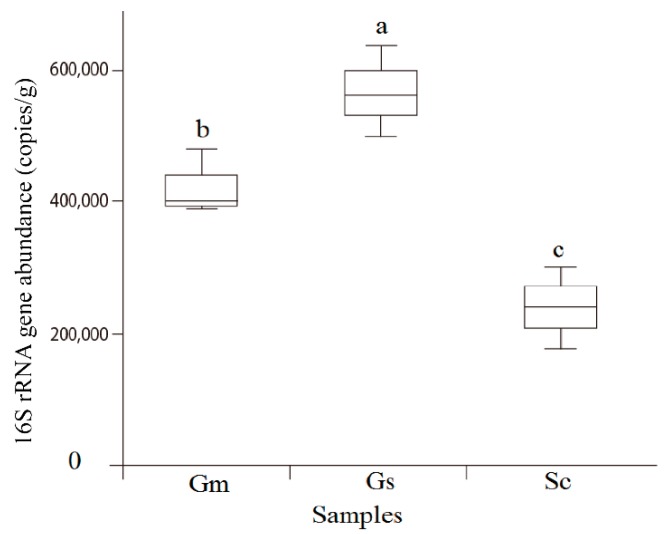
Quantitative PCR of the 16S rRNA gene abundance associated with *Sesbania cannabina* (Sc), *Glycine soja* (Gs), and *Glycine max* (Gm) nodules.

**Figure 3 microorganisms-08-00207-f003:**
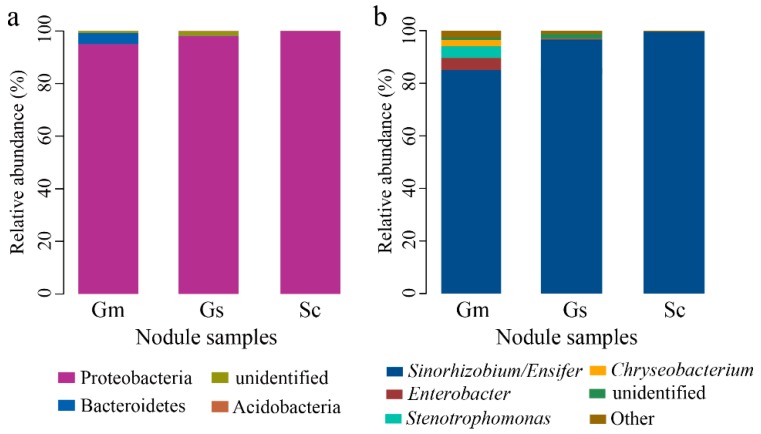
Relative abundances of bacteria at the phylum (**a**) and genus level (**b**) in nodule samples of *Sesbania cannabina* (Sc), *Glycine soja* (Gs), and *Glycine max* (Gm).

**Figure 4 microorganisms-08-00207-f004:**
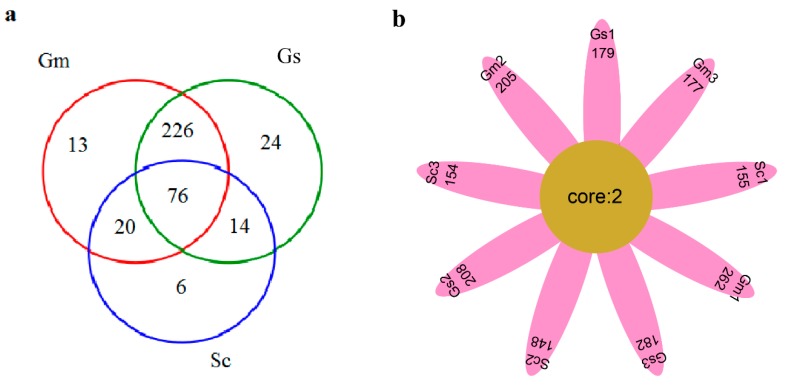
Venn diagram (**a**) and flower plot (**b**) showing the operational taxonomic units shared among different nodule samples associated with *Sesbania cannabina* (Sc), *Glycine soja* (Gs), and *Glycine max* (Gm).

**Figure 5 microorganisms-08-00207-f005:**
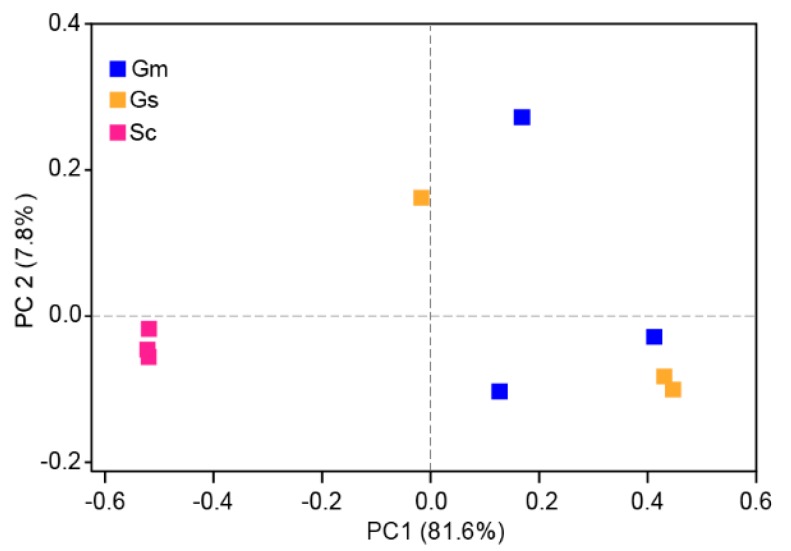
Principal coordinates analysis of the endophytic nodule bacterial communities in *Sesbania cannabina* (Sc), *Glycine soja* (Gs), and *Glycine max* (Gm) at the OTU level.

**Figure 6 microorganisms-08-00207-f006:**
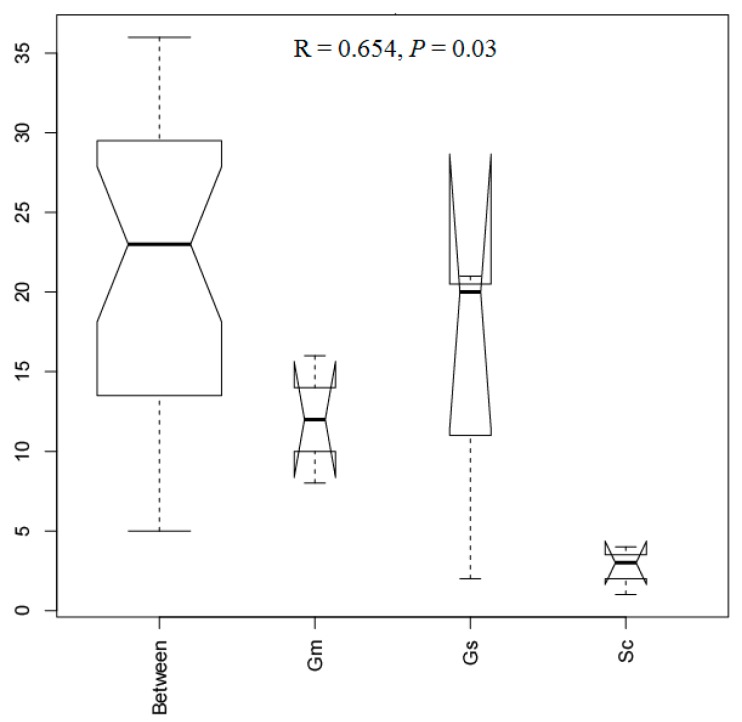
Results of an analysis of similarity (ANOSIM) test for differences among samples associated with *Sesbania cannabina* (Sc), *Glycine soja* (Gs), and *Glycine max* (Gm).

**Figure 7 microorganisms-08-00207-f007:**
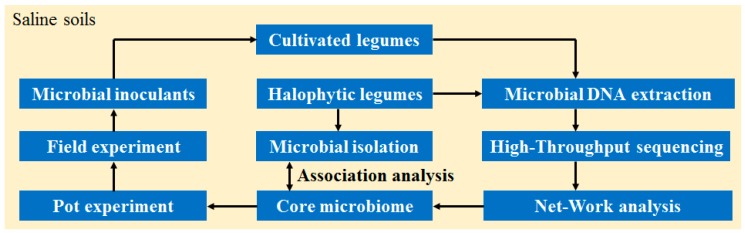
Conceptual strategy for the utilization of nodule microbes associated with halophytic legumes.

**Table 1 microorganisms-08-00207-t001:** Morphological characteristics and nitrogenase activity of the nodules of three legumes under saline soil conditions.

Plant Species	External Color of Nodules	Shape of Nodule	Nodule Size (mm)	Effective Nodule Weight Per Plant (g)	Nodule Nitrogenase Activity [μmol (g h)^−1^]
*G. max*	white, pink, brown	round to massive	3–9	0.39 ± 0.07 b	3.52 ± 0.15 b
*G. soja*	white, pink, brown	round to massive	3–11	0.57 ± 0.07 a	4.37 ± 0.17 a
*S. cannabina*	white, pink, brown	round	3–9	0.61 ± 0.05 a	4.27 ± 0.11 a

Values are the means of three replicates ± SD. Values within a column followed by different lowercase letters are significantly different (*p* < 0.05).

**Table 2 microorganisms-08-00207-t002:** Operational taxonomic unit (OTU) richness and diversity indices of different samples associated with examined legume nodules with a 97% similarity cutoff.

Species	OTUs	Coverage (%)	Chao1 Richness	Shannon Diversity	PD Whole Tree
*G. max*	217 ± 35 ab	87.6 ± 2.1 b	284 ± 41 a	5.78 ± 0.55 a	4.17 ± 0.77 a
*G. soja*	191 ± 13 a	90.2 ± 1.5 b	228 ± 30 a	5.88 ± 0.34 a	3.33 ± 0.62 ab
*S. cannabina*	154 ± 3 b	96.1 ± 0.2 a	112± 5 b	5.28 ± 0.04 a	1.82 ± 0.04 b

Values are the means of three replicates ± SD. Values within the same column followed by different lowercase letters are significantly different (*p* < 0.05).

**Table 3 microorganisms-08-00207-t003:** Sequences percentages of rhizobial and non-rhizobial species (top 40 abundant across all samples) associated with *Sesbania cannabina* (Sc), *Glycine soja* (Gs), and *Glycine max* (Gm) nodules. Since *Ensifer* and *Sinorhizobium* were synonymous genera, we used the name *Ensifer* to designate bacterial species in genus *Sinorhizobium* according to Young (2003) and Martens et al. (2008).

Species	Gm	Gs	Sc	Species	Gm	Gs	Sc
Rhizobia				*Ensifer saheli*	0.21	0.18	0.00
*Ensifer americanum*	42.81	46.45	0.56	*Ensifer kostiense*	0.31	0.06	0.00
*Ensifer alkalisoli* YIC4027	8.10	11.31	68.23	*Ensifer* sp. T1Gs6	0.18	0.10	0.00
*Ensifer* sp.	13.57	17.23	16.46	*Ensifer* sp. R7-568	0.13	0.14	0.00
*Ensifer* sp. MSMC310	5.99	7.18	0.00	*Ensifer* sp. 209	0.06	0.13	0.00
*Ensifer* sp. C9	0.90	0.87	7.59	Others	0.33	0.25	0.00
*Ensifer meliloti*	0.74	1.15	3.99	Total	85.80	95.62	100
*Ensifer* sp. CEQ1	2.84	2.64	0.00	**Non-rhizobial bacteria**			
*Ensifer fredii*	1.94	1.34	0.00	*Enterobacter cloacae*	3.62	0.13	0.00
*Ensifer* sp. ENCBTM 34	2.04	0.36	0.00	*Stenotrophomonas* sp. CanR-75	2.79	0.00	0.00
*Ensifer adhaerens*	0.82	1.51	0.00	*Stenotrophomonas maltophilia*	2.41	0.00	0.00
*Ensifer* sp. T2GRs3	0.89	0.75	0.53	unidentified	0.95	1.44	0.00
*Ensifer* sp. ORS 1236	0.62	0.57	0.21	*Chryseobacterium wanjuense*	0.01	1.35	0.00
*Ensifer numidicus*	0.13	0.39	0.79	*Flavobacterium johnsoniae*	1.15	0.02	0.00
*Ensifer* sp. L1GRs3	0.49	0.51	0.30	*Enterobacter asburiae*	0.83	0.00	0.00
*Ensifer* sp. SEMIA 6161	0.09	0.18	0.89	*Chryseobacterium* sp. KJ9C8	0.00	0.79	0.00
*Ensifer terangae*	1.00	0.13	0.00	*Sphingobacterium bambusae*	0.38	0.00	0.00
*Ensifer mexicanus*	0.41	0.60	0.00	*Xanthomonas pisi*	0.27	0.00	0.00
*Ensifer xinjiangense*	0.47	0.37	0.00	*Sphingobacterium* sp. 1.3	0.23	0.02	0.00
*Mesorhizobium mediterraneum*	0.07	0.17	0.45	*Stenotrophomonas* sp. 2TP1A	0.03	0.21	0.00
*Ensifer* sp. S39	0.33	0.33	0.00	*Enterobacter* sp. B1-2	0.20	0.00	0.00
*Ensifer* sp. ORS 1085	0.23	0.40	0.00	Others	1.33	0.42	0.00
*Ensifer kummerowiae*	0.10	0.32	0.00	Total	14.20	4.38	0.00

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
