# Peer review of "The Root Nodule Microbiome of Cultivated and Wild Halophytic Legumes Showed Similar Diversity but Distinct Community Structure in Yellow River Delta Saline Soils"

_microorganisms, 2020, doi:10.3390/microorganisms8020207_

Round 1

Reviewer 1 Report

This is a very interesting piece of work, well written and highly readable.

Major comments:

1) Introduction and Discussion should be shortened

2) Authors use the terms rhizobial/non-rhizobial and rhizobacteria/non-rhizobacteria as synonyms. This is really confusing. I suggest using rhizobial/non-rhizobial.

3) The English language should be revised accurately

Author Response

This is a very interesting piece of work, well written and highly readable.

Major comments:

Comments 1: 1) Introduction and Discussion should be shortened

Reply: Thanks for your suggestions. We have revised Introduction and Discussion sections thoroughly to make them more concise.

Comments 2: 2) Authors use the terms rhizobial/non-rhizobial and rhizobacteria/non-rhizobacteria as synonyms. This is really confusing. I suggest using rhizobial/non-rhizobial.

Reply: Thanks for your suggestions. We have changed to “rhizobial/non-rhizobial” throughout the manuscript.

Comments 3: 3) The English language should be revised accurately

Reply: Thanks for your comments. We have checked and revised the English language throughout the manuscript carefully.

Reviewer 2 Report

This is the review of the paper untitled ‘The root nodule microbiome of cultivated and wild halophytic legumes showed similar diversity but distinct community structure in Yellow River Delta saline soils’ by Yanfen Zheng et al submitted for publication in Microorganisms. The aim of the paper was to describe the microbial communities within nodules of 3 plant species. The sequencing strategy (PacBio) is very nice, allowing a high resolution in the sequence identification.

Despite the merits of the study, the paper has 2 major weaknesses which are related to each other :

1-Working hypotheses or null hypotheses are lacking in the paper. Thus when reading the introduction, it is very hard to understand why the work has been done, and on these hypotheses which decisions have been taken in terms of experiments

2-experimental design :

The study has been done on 3 plant species and 3 replicates for each. Thus, 9 samples in total are analyzed. There is no information about the soil used to grow the plants. For one given plant all the nodules were mixed altogether. Related to my previous point raised, It is very unclear why this experimental design. The fact that we don’t know what is the variance in the microbial community composition among nodules of a given plant is a weakness; the fact that we have a very limited idea about the variance in the microbial community composition among plants of a given species is also problematic with Glycine max displaying variability conversely to S cannabina. For one given plant species, because nodules were sampled from only 3 plants we don’t really know if the results are just a consequence of the sampling.

From these 2 comments, it would be needed to rewrite the introduction and have clear hypotheses tested from your experimental design. The experimental design has likely also to be rethink.

Other comments ‘Methods’ section

L 117-119 Remove the sentence which has nothing to see with the DNA extraction

L133 The full length → replace by the ‘The near full lenght’ or ‘The ~1465bp fragment’

L143 Please indicate the total number of sequence produced and the number of sequences per sample (a mean ±SD could be sufficient)

L160 Did you normalized the sequencing depth in your dataset across samples ? Aspect related to this is said later L 206 but without enough information

L162-164 Did you transformed the data before doing PCA ; In PCA, Euclidean distance to measure dissimilarity among pairs of communities are used. Thus your dataset to calculate the PCA contains many many zeroes, which I guess it is the case given your microbial community structure (few OTUs highly dominants and most of the others being rare). It is known that in this case PCA can produce severe artefacts. In this case, PcoA should be prefered and the distance matrix has to be calculated not from Euclidean distances.

L166 I am not fully familiar with the data analysis, but again here UPGMA have particular assumptions. One especially strong assumption under UPGMA is similar evolutionary tempo ( ie similar mutation rate per unit of time across all the sequence analyzed). I am not fully sure that this clustering method is the best. If it is the case you have to explain and argue your choice in the paragraph (ie why this choice) and to emphasize what for.

Other comments ‘Results’ section

Fig 1 you can provide the rarefaction curves for each sample. Alternatively, a rank-abundance curve for each sample

Table 2 results indicated in the table do not fit with results in Fig 1

Fig 2 legend on the Y axis is wrong

Fig 3 Please choose different colors among panels A and B

I did not comment on the Discussion section because it has to be in line with working hypotheses which are missing at this stage.

Author Response

This is the review of the paper untitled ‘The root nodule microbiome of cultivated and wild halophytic legumes showed similar diversity but distinct community structure in Yellow River Delta saline soils’ by Yanfen Zheng et al submitted for publication in Microorganisms. The aim of the paper was to describe the microbial communities within nodules of 3 plant species. The sequencing strategy (PacBio) is very nice, allowing a high resolution in the sequence identification.

Despite the merits of the study, the paper has 2 major weaknesses which are related to each other :

Comments 1: 1-Working hypotheses or null hypotheses are lacking in the paper. Thus when reading the introduction, it is very hard to understand why the work has been done, and on these hypotheses which decisions have been taken in terms of experiments

Reply: Thanks for your critical comments. We have revised introduction carefully and added hypotheses in Lines 59-61.

Comments 2: 2-experimental design :

The study has been done on 3 plant species and 3 replicates for each. Thus, 9 samples in total are analyzed. There is no information about the soil used to grow the plants. For one given plant all the nodules were mixed altogether. Related to my previous point raised, It is very unclear why this experimental design. The fact that we don’t know what is the variance in the microbial community composition among nodules of a given plant is a weakness; the fact that we have a very limited idea about the variance in the microbial community composition among plants of a given species is also problematic with Glycine max displaying variability conversely to S cannabina. For one given plant species, because nodules were sampled from only 3 plants we don’t really know if the results are just a consequence of the sampling.

Reply: Thanks for your valuable comments. (1) The chemical properties of soil used to grow the plants were showed in Lines 74-77. (2) We have added hypotheses at the end of Introduction section (Lines 59-61). In this study, we aimed to compare and characterize nodule microbiomes of different legumes at species level using PacBio sequencing. It could provide us deep insights to utilize microbial resources. The results showed that the dominant species of each plant species were different. Ensifer alkalisoli YIC4027 was dominated in S. cannabina (68.23%), while the dominant species in G. max was E. americanum (42.81%), as did in G. soja (46.45%). This result was showed in Lines 219-221. (3) Three sampling points with five plants (i.e. a total of 15 plants for each plant species) were collected (this information showed in Lines 83-84). Thus, we think our results are representative. (3) The dominant species of each plant species were different. Ensifer alkalisoli YIC4027 was dominated the dominated species in S. cannabina (68.23%), while the dominant species in G. max was E. americanum (42.81%), as well as did in G. soja (46.45%). (Lines 264-266)

Comments 3: From these 2 comments, it would be needed to rewrite the introduction and have clear hypotheses tested from your experimental design. The experimental design has likely also to be rethink.

Reply: Thanks for your constructive comments. We have rewritten the Introduction section thoroughly and raised working hypotheses.

Other comments ‘Methods’ section

Comments 4: L117-119 Remove the sentence which has nothing to see with the DNA extraction

Reply: We have removed the irrelevant sentences.

Comments 5: L133 The full length → replace by the ‘The near full lenght’ or ‘The ~1465bp fragment’

Reply: We have replaced ‘The full length’ by ‘The near full length’.

Comments 6: L143 Please indicate the total number of sequence produced and the number of sequences per sample (a mean ±SD could be sufficient)

Reply: Thanks for your suggestion. We have added this information in Lines 167-168.

Comments 7: L160 Did you normalized the sequencing depth in your dataset across samples? Aspect related to this is said later L 206 but without enough information

Reply: Thanks for your comments. We have added detail information in Lines 168-169.

Comments 8: L162-164 Did you transformed the data before doing PCA ; In PCA, Euclidean distance to measure dissimilarity among pairs of communities are used. Thus your dataset to calculate the PCA contains many many zeroes, which I guess it is the case given your microbial community structure (few OTUs highly dominants and most of the others being rare). It is known that in this case PCA can produce severe artefacts. In this case, PcoA should be prefered and the distance matrix has to be calculated not from Euclidean distances.

Reply: Thanks for your suggestions. We have transformed data and performed PCoA based on Bray-curtis distance. Please see new Figure 5.

Comments 9: L166 I am not fully familiar with the data analysis, but again here UPGMA have particular assumptions. One especially strong assumption under UPGMA is similar evolutionary tempo ( ie similar mutation rate per unit of time across all the sequence analyzed). I am not fully sure that this clustering method is the best. If it is the case you have to explain and argue your choice in the paragraph (ie why this choice) and to emphasize what for.

Reply: Sorry, it is a mistake. We didn’t do UPGMA clustering tree and we have removed these sentences from methods section.

Other comments ‘Results’ section

Comments 10: Fig 1 you can provide the rarefaction curves for each sample. Alternatively, a rank-abundance curve for each sample

Reply: Thanks for your suggestions. We have provided rarefaction curves for each sample (Figure 1).

Comments 11: Table 2 results indicated in the table do not fit with results in Fig 1

Reply: Sorry for our mistake. We have checked our data carefully and changed Figure 1 as well as Table 2.

Comments 12: Fig 2 legend on the Y axis is wrong

Reply: Sorry for our mistake. We have modified it.

Comments 13: Fig 3 Please choose different colors among panels A and B

Reply: Thanks for your suggestion. The colour in panel A has been changed.

Comments 14: I did not comment on the Discussion section because it has to be in line with working hypotheses which are missing at this stage.

Reply: Thanks for your suggestions. Discussion section has been revised thoroughly to be in line with hypotheses and it is more readable now.

Reviewer 3 Report

The manuscript presented by Zheng et al use PacBio sequencing to elucidate the differences in the bacterial microbiomes existing in the nodules of 2 wild and 1 cultivated halophytic legumes. The use of this high-throughput technique to obtain and compare community data from nodules is the main interest of the manuscript. However, comparing data between non-saline and saline conditions would be a plus. Nonetheless, the manuscript is fine, and I only have some minor comments:

1.-The main concern here is that the samples (1 to 3) of each legume are quite dissimilar, specially both Glycine, difference showed in Table S2 and Figure 5 PCA. Please, provide an explanation about this limitation.

2.- How many nodules were processed in each sample for PacBio sequencing? Please complete the data analysis section of the Methods with information about the processing of the raw data. Please, check Johnson et al 2019 (Johnson, J.S., Spakowicz, D.J., Hong, B. et al. Evaluation of 16S rRNA gene sequencing for species and strain-level microbiome analysis. Nat Commun 10, 5029 (2019) doi:10.1038/s41467-019-13036-1) in order to improve this section, if it suits you.

3.-Figure 1. The rarefaction curve of Sc has no error bars.

4.- Lines 76-77 Please, revise this sentence “…was to identify and characteristic the nodule microbiomes of…”

5.- Line 220. Please, clarify if you have used Greengenes or SILVA as you stated you used SILVA in the Methods section. Moreover, you should clarify which is the medium size of the reads.

6.- Ensifer sp YIC4027 is classified within Ensifer alkalisoli, according to Dang et al 2019 (Dang, X., Xie, Z., Liu, W. et al. The genome of Ensifer alkalisoli YIC4027 provides insights for host specificity and environmental adaptations. BMC Genomics 20, 643 (2019) doi:10.1186/s12864-019-6004-7), who also found evidences of adaptation to alkaline- saline soils in its genome.

Author Response

The manuscript presented by Zheng et al use PacBio sequencing to elucidate the differences in the bacterial microbiomes existing in the nodules of 2 wild and 1 cultivated halophytic legumes. The use of this high-throughput technique to obtain and compare community data from nodules is the main interest of the manuscript. However, comparing data between non-saline and saline conditions would be a plus. Nonetheless, the manuscript is fine, and I only have some minor comments:

Reply: Thanks for your valuable suggestions. That’s a good point. We will explore legumes’ microbiome under non-saline and saline conditions in the future study.

1.-The main concern here is that the samples (1 to 3) of each legume are quite dissimilar, specially both Glycine, difference showed in Table S2 and Figure 5 PCA. Please, provide an explanation about this limitation.

Reply: Thanks for your comments. We have discussed this limitation in Lines 312-315.

2.- How many nodules were processed in each sample for PacBio sequencing? Please complete the data analysis section of the Methods with information about the processing of the raw data. Please, check Johnson et al 2019 (Johnson, J.S., Spakowicz, D.J., Hong, B. et al. Evaluation of 16S rRNA gene sequencing for species and strain-level microbiome analysis. Nat Commun 10, 5029 (2019) doi:10.1038/s41467-019-13036-1) in order to improve this section, if it suits you.

Reply: Each sample contains 0.5 g nodules and this information has been added in Lines 101-102. We have rewritten data analysis section and added information about how to process raw data (Lines 134-138).

3.-Figure 1. The rarefaction curve of Sc has no error bars.

Reply: Thanks for your comments. We have provided a new figure that rarefaction curves of all nine samples are shown according to the suggestion of Reviewer 2.

4.- Lines 76-77 Please, revise this sentence “…was to identify and characteristic the nodule microbiomes of…”

Reply: Thanks for your comments. We have revised this sentence to “In order to characterize and compare their nodule microbiomes at the species level with high throughput, …”. Lines 64-65.

5.- Line 220. Please, clarify if you have used Greengenes or SILVA as you stated you used SILVA in the Methods section. Moreover, you should clarify which is the medium size of the reads.

Reply: Sorry for our omission. We used silva database to perform taxonomic annotation. This sentence has been modified. We have showed the number of reads per sample (a mean ±SD) in Lines 167-168.

6.- Ensifer sp YIC4027 is classified within Ensifer alkalisoli, according to Dang et al 2019 (Dang, X., Xie, Z., Liu, W. et al. The genome of Ensifer alkalisoli YIC4027 provides insights for host specificity and environmental adaptations. BMC Genomics 20, 643 (2019) doi:10.1186/s12864-019-6004-7), who also found evidences of adaptation to alkaline- saline soils in its genome.

Reply: Thanks for your comments. We have changed “Ensifer sp YIC4027” to “Ensifer alkalisoli YIC4027” throughout this manuscript and cited this paper in discussion.